# Graph Neural Network-Inspired Kernels for Gaussian Processes in Semi-Supervised Learning

**Zehao Niu**[1], **Mihai Anitescu**[1,2]
[1]University of Chicago, [2]Argonne National Laboratory
niuzehao@uchicago.edu
anitescu@mcs.anl.gov

**Jie Chen**[*]
MIT-IBM Watson AI Lab
IBM Research
chenjie@us.ibm.com

## Abstract

Gaussian processes (GPs) are an attractive class of machine learning models because of their simplicity and flexibility as building blocks of more complex Bayesian models. Meanwhile, graph neural networks (GNNs) emerged recently as a promising class of models for graph-structured data in semi-supervised learning and beyond. Their competitive performance is often attributed to a proper capturing of the graph inductive bias. In this work, we introduce this inductive bias into GPs to improve their predictive performance for graph-structured data. We show that a prominent example of GNNs, the graph convolutional network, is equivalent to some GP when its layers are infinitely wide; and we analyze the kernel universality and the limiting behavior in depth. We further present a programmable procedure to compose covariance kernels inspired by this equivalence and derive example kernels corresponding to several interesting members of the GNN family. We also propose a computationally efficient approximation of the covariance matrix for scalable posterior inference with large-scale data. We demonstrate that these graph-based kernels lead to competitive classification and regression performance, as well as advantages in computation time, compared with the respective GNNs.

## 1 Introduction

Gaussian processes (GPs) (Rasmussen & Williams, 2006) are widely used in machine learning, uncertainty quantification, and global optimization. In the Bayesian setting, a GP serves as a prior probability distribution over functions, characterized by a mean (often treated as zero for simplicity) and a covariance. Conditioned on observed data with a Gaussian likelihood, the random function admits a posterior distribution that is also Gaussian, whose mean is used for prediction and the variance serves as an uncertainty measure. The closed-form posterior allows for exact Bayesian inference, resulting in great attractiveness and wide usage of GPs.

The success of GPs in practice depends on two factors: the observations (training data) and the covariance kernel. We are interested in semi-supervised learning, where only a small amount of data is labeled while a large amount of unlabeled data can be used together for training (Zhu, 2008). In recent years, graph neural networks (GNNs) (Zhou et al., 2020; Wu et al., 2021) emerged as a promising class of models for this problem, when the labeled and unlabeled data are connected by a graph. The graph structure becomes an important inductive bias that leads to the success of GNNs. This inductive bias inspires us to design a GP model under limited observations, by building the graph structure into the covariance kernel.

An intimate relationship between neural networks and GPs is known: a neural network with fully connected layers, equipped with a prior probability distribution on the weights and biases, converges to a GP when each of its layers is infinitely wide (Lee et al., 2018; de G. Matthews et al., 2018). Such a result is owing to the central limit theorem (Neal, 1994; Williams, 1996) and the GP covariance

---

[*]To whom correspondence should be addressed.

can be recursively computed if the weights (and biases) in each layer are iid Gaussian. Similar results for other architectures, such as convolution layers and residual connections, were subsequently established in the literature (Novak et al., 2019; Garriga-Alonso et al., 2019).

One focus of this work is to establish a similar relationship between GNNs and the limiting GPs. We will derive the covariance kernel that incorporates the graph inductive bias as GNNs do. We start with one of the most widely studied GNNs, the graph convolutional network (GCN) (Kipf & Welling, 2017), and analyze the kernel universality as well as the limiting behavior when the depth also tends to infinity. We then derive covariance kernels from other GNNs by using a programmable procedure that corresponds every building block of a neural network to a kernel operation.

Meanwhile, we design efficient computational procedures for posterior inference (i.e., regression and classification). GPs are notoriously difficult to scale because of the cubic complexity with respect to the number of training data. Benchmark graph datasets used by the GNN literature may contain thousands or even millions of labeled nodes (Hu et al., 2020b). The semi-supervised setting worsens the scenario, as the covariance matrix needs to be (recursively) evaluated in full because of the graph convolution operation. We propose a Nyström-like scheme to perform low-rank approximations and apply the approximation recursively on each layer, to yield a low-rank kernel matrix. Such a matrix can be computed scalably. We demonstrate through numerical experiments that the GP posterior inference is much faster than training a GNN and subsequently performing predictions on the test set.

We summarize the contributions of this work as follows:

1. We derive the GP as a limit of the GCN when the layer widths tend to infinity and study the kernel universality and the limiting behavior in depth.
2. We propose a computational procedure to compute a low-rank approximation of the covariance matrix for practical and scalable posterior inference.
3. We present a programmable procedure to compose covariance kernels and their approximations and show examples corresponding to several interesting members of the GNN family.
4. We conduct comprehensive experiments to demonstrate that the GP model performs favorably compared to GNNs in prediction accuracy while being significantly faster in computation.

## 2 RELATED WORK

It has long been observed that GPs are limits of standard neural networks with one hidden layer when the layer width tends to infinity (Neal, 1994; Williams, 1996). Recently, renewed interests in the equivalence between GPs and neural networks were extended to deep neural networks (Lee et al., 2018; de G. Matthews et al., 2018) as well as modern neural network architectures, such as convolution layers (Novak et al., 2019), recurrent networks (Yang, 2019), and residual connections (Garriga-Alonso et al., 2019). The term NNGP (neural network Gaussian process) henceforth emerged under the context of Bayesian deep learning. Besides the fact that an infinite neural network defines a kernel, the training of a neural network by using gradient descent also defines a kernel—the neural tangent kernel (NTK)—that describes the evolution of the network (Jacot et al., 2018; Lee et al., 2019). Library supports in Python were developed to automatically construct the NNGP and NTK kernels based on programming the corresponding neural networks (Novak et al., 2020).

GNNs are neural networks that handle graph-structured data (Zhou et al., 2020; Wu et al., 2021). They are a promising class of models for semi-supervised learning. Many GNNs use the message-passing scheme (Gilmer et al., 2017), where neighborhood information is aggregated to update the representation of the center node. Representative examples include GCN (Kipf & Welling, 2017), GraphSAGE (Hamilton et al., 2017), GAT (Veličković et al., 2018), and GIN (Xu et al., 2019). It is found that the performance of GNNs degrades as they become deep; one approach to mitigating the problem is to insert residual/skip connections, as done by JumpingKnowledge (Xu et al., 2018), APPNP (Gasteiger et al., 2019), and GCNII (Chen et al., 2020).

GP inference is too costly, because it requires the inverse of the $N \times N$ dense kernel matrix. Scalable approaches include low-rank methods, such as Nyström approximation (Drineas & Mahoney, 2005), random features (Rahimi & Recht, 2007), and KISS-GP (Wilson & Nickisch, 2015); as well as multi-resolution (Katzfuss, 2017) and hierarchical methods (Chen et al., 2017; Chen & Stein, 2021).

Prior efforts on integrating graphs into GPs exist. Ng et al. (2018) define a GP kernel by combing a base kernel with the adjacency matrix; it is related to a special case of our kernels where the network has only one layer and the output admits a robust-max likelihood for classification. Hu et al. (2020a) explore a similar route to us, by taking the limit of a GCN, but its exploration is less comprehensive because it does not generalize to other GNNs and does not tackle the scalability challenge.

## 3 GRAPH CONVOLUTIONAL NETWORK AS A GAUSSIAN PROCESS

We start with a few notations used throughout this paper. Let an undirected graph be denoted by $\mathcal{G} = (\mathcal{V}, \mathcal{E})$ with $N = |\mathcal{V}|$ nodes and $M = |\mathcal{E}|$ edges. For notational simplicity, we use $A \in \mathbb{R}^{N \times N}$ to denote the original graph adjacency matrix or any modified/normalized version of it. Using $d_l$ to denote the width of the $l$-th layer, the layer architecture of GCN reads

$$X^{(l)} = \phi \left( A X^{(l-1)} W^{(l)} + b^{(l)} \right), \tag{1}$$

where $X^{(l-1)} \in \mathbb{R}^{N \times d_{l-1}}$ and $X^{(l)} \in \mathbb{R}^{N \times d_l}$ are layer inputs and outputs, respectively; $W^{(l)} \in \mathbb{R}^{d_{l-1} \times d_l}$ and $b^{(l)} \in \mathbb{R}^{1 \times d_l}$ are the weights and the biases, respectively; and $\phi$ is the ReLU activation function. The graph convolutional operator $A$ is a symmetric normalization of the graph adjacency matrix with self-loops added (Kipf & Welling, 2017).

For ease of exposition, it will be useful to rewrite the matrix notation (1) in element-sums and products. To this end, for a node $x$, let $z_i^{(l)}(x)$ and $x_i^{(l)}(x)$ denote the pre- and post-activation value at the $i$-th coordinate in the $l$-th layer, respectively. Particularly, in an $L$-layer GCN, $x^{(0)}(x)$ is the input feature vector and $z^{(L)}(x)$ is the output vector. The layer architecture of GCN reads

$$y_i^{(l)}(x) = \sum_{j=1}^{d_{l-1}} W_{ji}^{(l)} x_j^{(l-1)}(x), \quad z_i^{(l)}(x) = b_i^{(l)} + \sum_{v \in \mathcal{V}} A_{xv} y_i^{(l)}(v), \quad x_i^{(l)}(x) = \phi(z_i^{(l)}(x)). \tag{2}$$

### 3.1 LIMIT IN THE WIDTH

The following theorem states that when the weights and biases in each layer are iid zero-mean Gaussians, in the limit on the layer width, the GCN output $z^{(L)}(x)$ is a multi-output GP over the index $x$.

**Theorem 1.** *Assume $d_1, \ldots, d_{L-1}$ to be infinite in succession and let the bias and weight terms be independent with distributions*

$$b_i^{(l)} \sim \mathcal{N}(0, \sigma_b^2), \quad W_{ij}^{(l)} \sim \mathcal{N}(0, \sigma_w^2 / d_{l-1}), \quad l = 1, \ldots, L.$$

*Then, for each $i$, the collection $\{z_i^{(l)}(x)\}$ over all graph nodes $x$ follows the normal distribution $\mathcal{N}(0, K^{(l)})$, where the covariance matrix $K^{(l)}$ can be computed recursively by*

$$C^{(l)} = \mathrm{E}_{z_i^{(l)} \sim \mathcal{N}(0, K^{(l)})}[\phi(z_i^{(l)})\phi(z_i^{(l)})^T], \quad l = 1, \ldots, L, \tag{3}$$

$$K^{(l+1)} = \sigma_b^2 \mathbf{1}_{N \times N} + \sigma_w^2 A C^{(l)} A^T, \qquad l = 0, \ldots, L-1. \tag{4}$$

All proofs of this paper are given in the appendix. Note that different from a usual GP, which is a random function defined over a connected region of the Euclidean space, here $z^{(L)}$ is defined over a discrete set of graph nodes. In the usual use of a graph in machine learning, this set is finite, such that the function distribution degenerates to a multivariate distribution. In semi-supervised learning, the dimension of the distribution, $N$, is fixed when one conducts transductive learning; but it will vary in the inductive setting because the graph will have new nodes and edges. One special care of a graph-based GP over a usual GP is that the covariance matrix will need to be recomputed from scratch whenever the graph alters.

Theorem 1 leaves out the base definition $C^{(0)}$, whose entry denotes the covariance between two input nodes. The traditional literature uses the inner product $C^{(0)}(x, x') = \frac{x \cdot x'}{d_0}$ (Lee et al., 2018),

but nothing prevents us using any positive-definite kernel alternatively.[1] For example, we could use the squared exponential kernel $C^{(0)}(x, x') = \exp\left(-\frac{1}{2}\sum_{j=1}^{d_0}\left(\frac{x_j - x'_j}{\ell_j}\right)^2\right)$. Such flexibility in essence performs an implicit feature transformation as preprocessing.

## 3.2 UNIVERSALITY

A covariance kernel is positive definite; hence, the Moore–Aronszajn theorem (Aronszajn, 1950) suggests that it defines a unique Hilbert space for which it is a reproducing kernel. If this space is dense, then the kernel is called *universal*. One can verify universality by checking if the kernel matrix is positive definite for any set of distinct points.[2] For the case of graphs, it suffices to verify if the covariance matrix for all nodes is positive definite.

We do this job for the ReLU activation function. It is known that the kernel $\mathrm{E}_{w\sim\mathcal{N}(0,I_d)}[\phi(w \cdot x)\phi(w \cdot x')]$ admits a closed-form expression as a function of the angle between $x$ and $x'$, hence named the *arc-cosine* kernel (Cho & Saul, 2009). We first establish the following lemma that states that the kernel is universal over a half-space.

**Lemma 2.** *The arc-cosine kernel is universal on the upper-hemisphere $S = \left\{x \in \mathbb{R}^d : \|x\|_2 = 1, x_1 > 0\right\}$ for all $d \geq 2$.*

It is also known that the expectation in (3) is proportional to the arc-cosine kernel up to a factor $\sqrt{K^{(l)}(x,x)K^{(l)}(x',x')}$ (Lee et al., 2018). Therefore, we iteratively work on the post-activation covariance (3) and the pre-activation covariance (4) and show that the covariance kernel resulting from the limiting GCN is universal, for any GCN with three or more layers.

**Theorem 3.** *Assume $A$ is irreducible and non-negative and $C^{(0)}$ does not contain two linearly dependent rows. Then, $K^{(l)}$ is positive definite for all $l \geq 3$.*

## 3.3 LIMIT IN THE DEPTH

The depth of a neural network exhibits interesting behaviors. Deep learning tends to favor deep networks because of their empirically outstanding performance, exemplified by generations of convolutional networks for the ImageNet classification (Krizhevsky et al., 2012; Wortsman et al., 2022); while graph neural networks are instrumental to be shallow because of the over-smoothing and over-squashing properties (Li et al., 2018; Topping et al., 2022). For multi-layer perceptrons (networks with fully connected layers), several previous works have noted that the recurrence relation of the covariance kernel across layers leads to convergence to a fixed-point kernel, when the depth $L \to \infty$ (see, e.g., Lee et al. (2018); in Appendix B.5, we elaborate this limit). In what follows, we offer the parallel analysis for GCN.

**Theorem 4.** *Assume $A$ is symmetric, irreducible, aperiodic, and non-negative with Perron-Frobenius eigenvalue $\lambda > 0$. The following results hold as $l \to \infty$.*

1. *When $\sigma_b^2 = 0$, $\rho_{\min}(K^{(l)}) \nearrow 1$, where $\rho_{\min}$ denotes the minimum correlation between any two nodes $x$ and $x'$.*

2. *When $\sigma_w^2 < 2/\lambda^2$, a subsequence of $K^{(l)}$ converges to some matrix.*

3. *When $\sigma_w^2 > 2/\lambda^2$, let $c_l = (\sigma_w^2\lambda^2/2)^l$; then, $K^{(l)}/c_l \to vv^T$ where $v$ is an eigenvector corresponding to $\lambda$.*

A few remarks follow. The first case implies that the correlation matrix converges monotonously to a matrix of all ones. As a consequence, up to some scaling $c'_l$ that may depend on $l$, the scaled covariance matrix $K^{(l)}/c'_l$ converges to a rank-1 matrix. The third case shares a similar result, with the limit explicitly spelled out, but note that the eigenvector $v$ may not be normalized. The second case is challenging to analyze. According to empirical verification, we speculate a stronger result—convergence of $K^{(l)}$ to a unique fixed point—may hold.

---

[1] Here, we abuse the notation and use $x$ in place of $x^{(0)}(x)$ in the inner product.

[2] Note the conventional confusion in terminology between functions and matrices: a kernel function is positive definite (resp. strictly positive definite) if the corresponding kernel matrix is positive semi-definite (resp. positive definite) for any collection of distinct points.

Table 1: Computational costs. $M$: number of edges; $N$: number of nodes; $N_b$: number of training nodes; $N_*$: number of prediction nodes; $N_a$: number of landmark nodes; $L$: number of layers. Assume $N_b \gg N_a$. For posterior variance, assume only the diagonal is needed.

|  | Time $O(\cdot)$ | Storage $O(\cdot)$ |
| --- | --- | --- |
| Computation of $Q^{(L)}$ | $LMN_a + LNN_a^2 + LN_a^3$ | $NN_a$ |
| Posterior mean (6) | $N_*N_a + N_bN_a^2 + N_a^3$ | $(N_b + N_*)N_a$ |
| Posterior variance (7) | $N_*N_a + N_bN_a^2 + N_a^3$ | $(N_b + N_*)N_a$ |

## 4    SCALABLE COMPUTATION THROUGH LOW-RANK APPROXIMATION

The computation of the covariance matrix $K^{(L)}$ through recursion (3)–(4) is the main computational bottleneck for GP posterior inference. We start the exposition with the mean prediction. We compute the posterior mean $\widehat{y}_* = K^{(L)}_{*b}(K^{(L)}_{bb} + \epsilon I)^{-1}y_b$, where the subscripts $b$ and $*$ denote the training set and the prediction set, respectively; and $\epsilon$, called the *nugget*, is the noise variance of the training data. Let there be $N_b$ training nodes and $N_*$ prediction nodes. It is tempting to compute only the $(N_b + N_*) \times N_b$ submatrix of $K^{(L)}$ for the task, but the recursion (4) requires the full $C^{(L-1)}$ at the presence of $A$, and hence all the full $C^{(l)}$'s and $K^{(l)}$'s.

To reduce the computational costs, we resort to a low-rank approximation of $C^{(l)}$, from which we easily see that $K^{(l+1)}$ is also low-rank. Before deriving the approximation recursion, we note (again) that for the ReLU activation $\phi$, $C^{(l)}$ in (3) is the arc-cosine kernel with a closed-form expression:

$$C^{(l)}_{xx'} = \frac{1}{2\pi}\sqrt{K^{(l)}_{xx}K^{(l)}_{x'x'}}\left(\sin\theta^{(l)}_{xx'} + (\pi - \theta^{(l)}_{xx'})\cos\theta^{(l)}_{xx'}\right) \quad \text{where} \quad \theta^{(l)}_{xx'} = \arccos\left(\frac{K^{(l)}_{xx'}}{\sqrt{K^{(l)}_{xx}K^{(l)}_{x'x'}}}\right).$$
(5)

Hence, the main idea is that starting with a low-rank approximation of $K^{(l)}$, compute an approximation of $C^{(l)}$ by using (5), and then obtain an approximation of $K^{(l+1)}$ based on (4); then, repeat.

To derive the approximation, we use the subscript $a$ to denote a set of landmark nodes with cardinality $N_a$. The Nyström approximation (Drineas & Mahoney, 2005) of $K^{(0)}$ is $K^{(0)}_{:a}(K^{(0)}_{aa})^{-1}K^{(0)}_{a:}$, where the subscript : denotes retaining all rows/columns. We rewrite this approximation in the Cholesky style as $Q^{(0)}Q^{(0)^T}$, where $Q^{(0)} = K^{(0)}_{:a}(K^{(0)}_{aa})^{-\frac{1}{2}}$ has size $N \times N_a$. Proceed with induction. Let $K^{(l)}$ be approximated by $\widehat{K}^{(l)} = Q^{(l)}Q^{(l)^T}$, where $Q^{(l)}$ has size $N \times (N_a + 1)$. We apply (5) to compute an approximation to $C^{(l)}_{:a}$, namely $\widehat{C}^{(l)}_{:a}$, by using $\widehat{K}^{(l)}_{:a}$. Then, (4) leads to a Cholesky style approximation of $K^{(l+1)}$:

$$\widehat{K}^{(l+1)} = \sigma_b^2 \mathbf{1}_{N \times N} + \sigma_w^2 A\widehat{C}^{(l)}A^T \equiv Q^{(l+1)}Q^{(l+1)^T},$$

where $Q^{(l+1)} = \begin{bmatrix} \sigma_w A\widehat{C}^{(l)}_{:a}(\widehat{C}^{(l)}_{aa})^{-\frac{1}{2}} & \sigma_b \mathbf{1}_{N \times 1} \end{bmatrix}$. Clearly, $Q^{(l+1)}$ has size $N \times (N_a+1)$, completing the induction.

In summary, $K^{(L)}$ is approximated by a rank-$(N_a+1)$ matrix $\widehat{K}^{(L)} = Q^{(L)}Q^{(L)^T}$. The computation of $Q^{(L)}$ is summarized in Algorithm 1. Once it is formed, the posterior mean is computed as

$$\widehat{y}_* \approx \widehat{K}^{(L)}_{*b}(\widehat{K}^{(L)}_{bb} + \epsilon I)^{-1}y_b = Q^{(L)}_{*:}\left(Q^{(L)^T}_{b:}Q^{(L)}_{b:} + \epsilon I\right)^{-1}Q^{(L)^T}_{b:}y_b,$$
(6)

where note that the matrix to be inverted has size $(N_a + 1) \times (N_a + 1)$, which is assumed to be significantly smaller than $N_b \times N_b$. Similarly, the posterior variance is

$$\widehat{K}^{(L)}_{**} - \widehat{K}^{(L)}_{*b}(\widehat{K}^{(L)}_{bb} + \epsilon I)^{-1}\widehat{K}^{(L)}_{b*} = \epsilon Q^{(L)}_{*:}\left(Q^{(L)^T}_{b:}Q^{(L)}_{b:} + \epsilon I\right)^{-1}Q^{(L)^T}_{*:}.$$
(7)

The computational costs of $Q^{(L)}$ and the posterior inference (6)–(7) are summarized in Table 1

---

**Algorithm 1** Computing $K^{(L)} \approx \widehat{K}^{(L)} = Q^{(L)}Q^{(L)^T}$

---

**Require:** $Q^{(0)}$ such that $K^{(0)} \approx Q^{(0)}Q^{(0)^T}$
 1: **for** $l = 0, \dots, L-1$ **do**
 2:     Compute $\widehat{K}_{:a}^{(l)} = Q_{::}^{(0)}Q_{a:}^{(0)^T}$
 3:     Compute $\widehat{C}_{:a}^{(l)}$ by (5), where $C^{(l)}$ (resp. $K^{(l)}$) entries are replaced by $\widehat{C}^{(l)}$ (resp. $\widehat{K}^{(l)}$) entries
 4:     Compute $Q^{(l+1)} = \left[\sigma_w A\widehat{C}_{:a}^{(l)}(\widehat{C}_{aa}^{(l)})^{-\frac{1}{2}} \quad \sigma_b \mathbf{1}_{N \times 1}\right]$
 5: **end for**

---

Table 2: Neural network building blocks, kernel operations, and the low-rank counterpart.

| Building block | Neural network | Kernel operation | Low-rank operation |
|---|---|---|---|
| Input | $X \leftarrow X^{(0)}$ | $K \leftarrow C^{(0)}$ | $Q \leftarrow \mathrm{Chol}(C^{(0)})$ |
| Bias term | $X \leftarrow X + b$ | $K \leftarrow K + \sigma_b^2 \mathbf{1}_{N \times N}$ | $Q \leftarrow [Q \quad \sigma_b \mathbf{1}_{N \times 1}]$ |
| Weight term | $X \leftarrow XW$ | $K \leftarrow \sigma_w^2 K$ | $Q \leftarrow \sigma_w Q$ |
| Mixed weight term | $X \leftarrow X(\alpha I + \beta W)$ | $K \leftarrow (\alpha^2 + \beta^2 \sigma_w^2)K$ | $Q \leftarrow \sqrt{\alpha^2 + \beta^2 \sigma_w^2}Q$ |
| Graph convolution | $X \leftarrow AX$ | $K \leftarrow AKA^T$ | $Q \leftarrow AQ$ |
| Activation | $X \leftarrow \phi(X)$ | $K \leftarrow g(K)$ | $Q \leftarrow \mathrm{Chol}(g(QQ^T))$ |
| Independent addition | $X \leftarrow X_1 + X_2$ | $K \leftarrow K_1 + K_2$ | $Q \leftarrow [Q_1 \quad Q_2]$ |

## 5 COMPOSING GRAPH NEURAL NETWORK-INSPIRED KERNELS

Theorem 1, together with its proof, suggests that the covariance matrix of the limiting GP can be computed in a composable manner. Moreover, the derivation of Algorithm 1 indicates that the low-rank approximation of the covariance matrix can be similarly composed. Altogether, such a nice property allows one to easily derive the corresponding covariance matrix and its approximation for a new GNN architecture, like writing a program and obtaining a transformation of it automatically through operator overloading (Novak et al., 2020): the covariance matrix is a transformation of the GNN and the composition of the former is in exactly the same manner and order as that of the latter. We call the covariance matrices *programmable*.

For example, we write a GCN layer as $X \leftarrow A\phi(X)W + b$, where for notational simplicity, $X$ denotes pre-activation rather than post-activation as in the earlier sections. The activation $\phi$ on $X$ results in a transformation of the kernel matrix $K$ into $g(K)$, defined as:

$$g(K) := C = \mathrm{E}_{z \sim \mathcal{N}(0,K)}[\phi(z)\phi(z)^T], \tag{8}$$

due to (3). Moreover, if $K$ admits a low-rank approximation $QQ^T$, then $g(K)$ admits a low-rank approximation $PP^T$ where $P = \mathrm{Chol}(g(K))$ with

$$\mathrm{Chol}(C) := C_{:a}C_{aa}^{-\frac{1}{2}}.$$

The next operation—graph convolution—multiplies $A$ to the left of the post-activation. Correspondingly, the covariance matrix $K$ is transformed to $AKA^T$ and the low-rank approximation factor $Q$ is transformed to $AQ$. Then, the operation—multiplying the weight matrix $W$ to the right—will transform $K$ to $\sigma_w^2 K$ and $Q$ to $\sigma_w Q$. Finally, adding the bias $b$ will transform $K$ to $K + \sigma_b^2 \mathbf{1}_{N \times N}$ and $Q$ to $[Q \quad \sigma_b \mathbf{1}_{N \times 1}]$. Altogether, we have obtained the following updates per layer:

$$\begin{aligned} \mathrm{GCN}: \quad & X \leftarrow A\phi(X)W + b \\ & K \leftarrow \sigma_w^2 Ag(K)A^T + \sigma_b^2 \mathbf{1}_{N \times N} \\ & Q \leftarrow \left[\sigma_w A\,\mathrm{Chol}(g(QQ^T)) \quad \sigma_b \mathbf{1}_{N \times 1}\right]. \end{aligned}$$

One may verify the $K$ update against (3)–(4) and the $Q$ update against Algorithm 1. Both updates can be automatically derived based on the update of $X$.

We summarize the building blocks of a GNN and the corresponding kernel/low-rank operations in Table 2. The independent-addition building block is applicable to skip/residual connections. For

example, here is the composition for the GCNII layer (Chen et al., 2020) without a bias term, where a skip connection with $X^{(0)}$ occurs:

$$\text{GCNII}: \quad X \leftarrow \left( (1-\alpha)A\phi(X) + \alpha X^{(0)} \right) ((1-\beta)I + \beta W)$$

$$K \leftarrow \left( (1-\alpha)^2 Ag(K)A^T + \alpha^2 K^{(0)} \right) ((1-\beta)^2 + \beta^2 \sigma_w^2)$$

$$Q \leftarrow \begin{bmatrix} (1-\alpha)A\operatorname{Chol}(g(QQ^T)) & \alpha Q^{(0)} \end{bmatrix} \sqrt{(1-\beta)^2 + \beta^2 \sigma_w^2}.$$

For another example of the composability, we consider the popular GIN layer (Xu et al., 2019), which we assume uses a 2-layer MLP after the neighborhood aggregation:

$$\text{GIN}: \quad X \leftarrow \phi(A\phi(X)W + b)W' + b'$$

$$K \leftarrow \sigma_w^2 g(B) + \sigma_b^2 \mathbf{1}_{N \times N} \quad \text{where} \quad B = \sigma_w^2 Ag(K)A^T + \sigma_b^2 \mathbf{1}_{N \times N}$$

$$Q \leftarrow \begin{bmatrix} \sigma_w \operatorname{Chol}(g(PP^T)) & \sigma_b \mathbf{1}_{N \times 1} \end{bmatrix} \quad \text{where} \quad P = \begin{bmatrix} \sigma_w A \operatorname{Chol}(g(QQ^T)) & \sigma_b \mathbf{1}_{N \times 1} \end{bmatrix}.$$

Additionally, the updates for a GraphSAGE layer (Hamilton et al., 2017) are given in Appendix C.

## 6 EXPERIMENTS

In this section, we conduct a comprehensive set of experiments to evaluate the performance of the GP kernels derived by taking limits on the layer width of GCN and other GNNs. We demonstrate that these GPs are comparable with GNNs in prediction performance, while being significantly faster to compute. We also show that the low-rank version scales favorably, suitable for practical use.

**Datasets.** The experiments are conducted on several benchmark datasets of varying sizes, covering both classification and regression. They include predicting the topic of scientific papers organized in a citation network (Cora, Citeseer, PubMed, and ArXiv); predicting the community of online posts based on user comments (Reddit), and predicting the average daily traffic of Wikipedia pages using hyperlinks among them (Chameleon, Squirrel, and Crocodile). Details of the datasets (including sources and preprocessing) are given in Appendix D.

**Experiment environment, training details, and hyperparameters** are given in Appendix D.

**Prediction Performance: GCN-based comparison.** We first conduct the semi-supervised learning tasks on all datasets by using GCN and GPs with different kernels. These kernels include the one equivalent to the limiting GCN (GCNGP), a usual squared-exponential kernel (RBF), and the GGP kernel proposed by Ng et al. (2018).[3] Each of these kernels has a low-rank version (suffixed with -X). RBF-X and GGP-X[4] use the Nyström approximation, consistent with GCNGP-X.

GPs are by nature suitable for regression. For classification tasks, we use the one-hot representation of labels to set up a multi-output regression. Then, we take the coordinate with the largest output as the class prediction. Such an ad hoc treatment is widely used in the literature, as other more principled approaches (such as using the Laplace approximation on the non-Gaussian posterior) are too time-consuming for large datasets, meanwhile producing no noticeable gain in accuracy.

Table 3 summarizes the accuracy for classification and the coefficient of determination, $R^2$, for regression. Whenever randomness is involved, the performance is reported as an average over five runs. The results of the two tasks show different patterns. For classification, GCNGP(-X) is sligtly better than GCN and GGP(-X), while RBF(-X) is significantly worse than all others; moreover, the low-rank version is outperformed by using the full kernel matrix. On the other hand, for regression, GCNGP(-X) significantly outperforms GCN, RBF(-X), and GGP(-X); and the low-rank version becomes better. The less competitive performance of RBF(-X) is expected, as it does not leverage the graph inductive bias. It is attractive that GCNGP(-X) is competitive with GCN.

**Prediction Performance: Comparison with other GNNs.** In addition to GCN, we conduct experiments with several popularly used GNN architectures (GCNII, GIN, and GraphSAGE) and GPs

---

[3]We apply only the kernel but not the likelihood nor the variational inference used in Ng et al. (2018), for reasons given in Appendix D.

[4]GGP-X in our notation is the Nyström approximation of the GGP kernel, different from a technique under the same name in Ng et al. (2018), which uses additionally the validation set to compute the prediction loss.

Table 3: Performance of GCNGP, in comparison with GCN and typical GP kernels. The Micro-F1 score is reported for classification tasks and $R^2$ is reported for regression tasks.

|  | GCN | GCNGP | GCNGP-X | RBF | RBF-X | GGP | GGP-X |
|---|---|---|---|---|---|---|---|
| Cora | $0.8183_{\pm 0.0055}$ | **0.8280** | 0.7980 | 0.5860 | 0.5850 | 0.7850 | 0.7410 |
| Citeseer | $0.6941_{\pm 0.0079}$ | **0.7090** | 0.7080 | 0.6120 | 0.6090 | 0.7060 | 0.6470 |
| PubMed | $0.7649_{\pm 0.0058}$ | **0.7960** | 0.7810 | 0.7360 | 0.7340 | 0.7820 | 0.7380 |
| ArXiv | $0.6990_{\pm 0.0014}$ | OOM | **$0.7011_{\pm 0.0011}$** | OOM | 0.5382 | OOM | 0.6527 |
| Reddit | $0.9330_{\pm 0.0006}$ | OOM | **$0.9465_{\pm 0.0003}$** | OOM | 0.5920 | OOM | 0.9058 |
| Chameleon | $0.5690_{\pm 0.0376}$ | 0.6720 | **0.6852** | 0.5554 | 0.5613 | 0.5280 | 0.5311 |
| Squirrel | $0.4243_{\pm 0.0393}$ | 0.4926 | **0.4998** | 0.3187 | 0.3185 | 0.2440 | 0.2251 |
| Crocodile | $0.6976_{\pm 0.0323}$ | 0.8002 | **0.8013** | 0.6643 | 0.6710 | 0.6952 | 0.6810 |

Table 4: Performance comparison (Micro-F1) between GNNs and the corresponding GP kernels.

| architecture | PubMed | | ArXiv | | Reddit | |
|---|---|---|---|---|---|---|
|  | GNN | GNNGP | GNN | GNNGP-X | GNN | GNNGP-X |
| GCN | $0.7649_{\pm 0.0058}$ | 0.7960 | $0.6989_{\pm 0.0016}$ | $0.7011_{\pm 0.0011}$ | $0.9330_{\pm 0.0006}$ | $0.9465_{\pm 0.0003}$ |
| GCNII | $0.7558_{\pm 0.0096}$ | 0.7840 | $0.7008_{\pm 0.0021}$ | $0.6955_{\pm 0.0011}$ | $0.9482_{\pm 0.0007}$ | $0.9500_{\pm 0.0003}$ |
| GIN | $0.7406_{\pm 0.0112}$ | 0.7690 | $0.6340_{\pm 0.0056}$ | $0.6652_{\pm 0.0012}$ | $0.9398_{\pm 0.0016}$ | $0.9428_{\pm 0.0005}$ |
| GraphSAGE | $0.7535_{\pm 0.0047}$ | 0.7900 | $0.6984_{\pm 0.0021}$ | $0.6962_{\pm 0.0007}$ | $0.9628_{\pm 0.0007}$ | $0.9539_{\pm 0.0003}$ |

with the corresponding kernels. We test with the three largest datasets: PubMed, ArXiv, and Reddit, for the latter two of which a low-rank version of the GPs is used for computational feasibility.

Table 4 summarizes the results. The observations on other GNNs extend similarly those on the GCN. In particular, on PubMed the GPs noticeably improve over the corresponding GNNs, while on ArXiv and Reddit the two families perform rather similarly. An exception is GIN for ArXiv, which significantly underperforms the GP counterpart, as well as other GNNs. It may improve with an extensive hyperparameter tuning.

**Running time.** We compare the running time of the methods covered by Table 3. Different from usual neural networks, the training and inference of GNNs do not decouple in full-batch training. Moreover, there is not a universally agreed split between the training and the inference steps in GPs. Hence, we compare the total time for each method.

Figure 1 plots the timing results, normalized against the GCN time for ease of comparison. It suggests that GCNGP(-X) is generally faster than GCN. Note that the vertical axis is in the logarithmic scale. Hence, for some of the datasets, the speedup is even one to two orders of magnitude.

**Scalability.** For graphs, especially under the semi-supervised learning setting, the computational cost of a GP is much more complex than that of a usual one (which can be simply described as "cubic in the training set size"). One sees in Table 1 the many factors that determine the cost of our graph-based low-rank kernels. To explore the practicality of the proposed method, we use the timings gathered for Figure 1 to obtain an empirical scaling with respect to the graph size, $M + N$.

Figure 2 fits the running times, plotted in the log-log scale, by using a straight line. We see that for neither GCN nor GCNGP(-X), the actual running time closely follows a polynomial complexity. However, interestingly, the least-squares fittings all lead to a slope approximately 1, which agrees with a linear cost. Theoretically, only GCNGP-X and GCN are approximately linear with respect to $M + N$, while GCNGP is cubic.

**Analysis on the depth.** The performance of GCN deteriorates with more layers, known as the oversmoothing phenomenon. Adding residual/skip connections mitigates the problem, such as in GCNII. A natural question asks if the corresponding GP kernels behave similarly.

Figure 3 shows that the trends of GCN and GCNII are indeed as expected. Interestingly, their GP counterparts both remain stable for depth $L$ as large as 12. Our depth analysis (Theorem 4) suggests that in the limit, the GPs may perform less well because the kernel matrix may degenerate to rank 1. This empirical result indicates that the drop in performance may have not started yet.

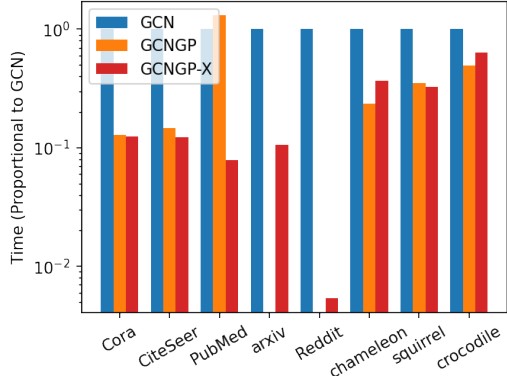

Figure 1: Timing comparison. For each dataset, the times are normalized against that of GCN.

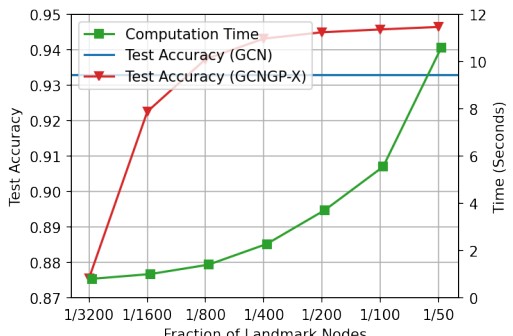

Figure 2: Scaling of the running time with respect to the graph size, $M + N$.

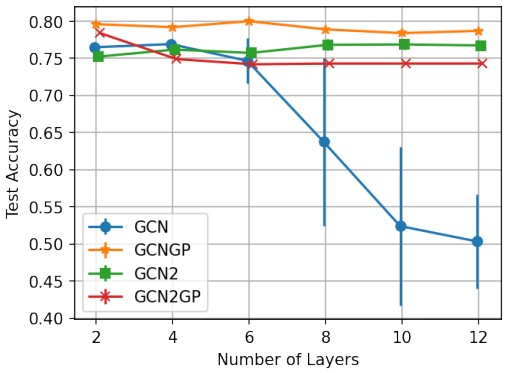

Figure 3: Performance of GNNs and GNNGPs as depth $L$ increases. Dataset: Pubmed.

Figure 4: Performance of GCNGP-X as number of landmark nodes increases. Dataset: Reddit.

**Analysis on the landmark set.** The number of landmark nodes, $N_a$, controls the approximation quality of the low-rank kernels and hence the prediciton accuracy. On the other hand, the computational costs summarized in Table 1 indicate a dependency on $N_a$ as high as the third power. It is crucial to develop an empirical understanding of the accuracy-time trade-off it incurs.

Figure 4 clearly shows that as $N_a$ becomes larger, the running time increase is not linear, while the increase of accuracy diminishes as the landmark set approaches the training set. It is remarkable that using only 1/800 of the training set as landmark nodes already achieves an accuracy surpassing that of GCN, by using time that is only a tiny fraction of the time otherwise needed to gain an additional 1% increase in the accuracy.

## 7 CONCLUSIONS

We have presented a GP approach for semi-supervised learning on graph-structured data, where the covariance kernel incorporates a graph inductive bias by exploiting the relationship between a GP and a GNN with infinitely wide layers. Similar to other neural networks priorly investigated, one can work out the equivalent GP (in particular, the covariance kernel) for GCN; and inspired by this equivalence, we formulate a procedure to compose covariance kernels corresponding to many other members of the GNN family. Moreover, every building block in the procedure has a low-rank counterpart, which allows for building a low-rank approximation of the covariance matrix that facilitates scalable posterior inference. We demonstrate the effectiveness of the derived kernels used for semi-supervised learning and show their advantages in computation time over GNNs.

## ACKNOWLEDGMENTS

Mihai Anitescu was supported by the U.S. Department of Energy, Office of Science, Office of Advanced Scientific Computing Research (ASCR) under Contract DE-AC02-06CH11347. Jie Chen acknowledges supports from the MIT-IBM Watson AI Lab.

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

# A CODE

Code is available at https://github.com/niuzehao/gnn-gp.

# B PROOFS AND ADDITIONAL THEOREMS

## B.1 PROOF OF THEOREM 1

We use mathematical induction. First, write the matrix form of (2) as follows:

$$\underset{N \times d_l}{Y^{(l)}} = \underset{N \times d_{l-1}}{X^{(l-1)}} \underset{d_{l-1} \times d_l}{W^{(l)}}, \quad \underset{N \times d_l}{Z^{(l)}} = \underset{N \times 1}{\mathbf{1}} \underset{1 \times d_l}{b^{(l)}} + \underset{N \times N}{A} \underset{N \times d_l}{Y^{(l)}}, \quad \underset{N \times d_l}{X^{(l)}} = \underset{N \times d_l}{\phi(Z^{(l)})}.$$

Assume each column of $Z^{(l-1)}$ is iid, then $X^{(l-1)}$ also has iid columns. Hence, $y_i^{(l)}(x)$ is a sum of iid terms and thus normal. Also, $(y_i^{(l)}(x_1), \cdots, y_i^{(l)}(x_N))$ are jointly multivariate normal, and identically distributed for different $i$; so they form a GP.

Moreover, terms like $y_i^{(l)}(x)$ and $y_{i'}^{(l)}(x')$ with $i \neq i'$ are jointly Gaussian, with zero covariance:

$$\mathrm{Cov}(y_i^{(l)}(x), y_{i'}^{(l)}(x')) = \sum_{j=1}^{d_{l-1}} \sum_{j'=1}^{d_{l-1}} \mathrm{Cov}\left(W_{ji}^{(l)} x_j^{(l-1)}(x), W_{j'i'}^{(l)} x_{j'}^{(l-1)}(x')\right) = 0.$$

Thus, they are independent, despite the fact that they may share the same $X^{(l-1)}$ terms. In conclusion, each column of $Y^{(l)}$ is an iid GP. Hence, each column of $Z^{(l)}$ is also an iid GP $\mathcal{N}(0, K^{(l)})$.

The covariance $K^{(l)}$ can be computed recursively. Similar to the case of a fully connected network, the covariance of $Y^{(l)}$ is

$$\mathrm{E}[y_i^{(l)}(x)y_i^{(l)}(x')] = \sigma_w^2 \mathrm{E}[x_j^{(l-1)}(x)x_j^{(l-1)}(x')] = \sigma_w^2 C^{(l-1)}(x, x'),$$

where

$$C^{(l-1)}(x, x') = \mathrm{E}_{z \sim \mathcal{N}(0, K^{(l-1)})}[\phi(z_j(x))\phi(z_j(x'))] \quad \text{for any } j.$$

In the matrix form, this is to say $\mathrm{Cov}(Y_j^{(l)}) = \sigma_w^2 \mathrm{E}[X_j^{(l-1)} X_j^{(l-1)^T}] = \sigma_w^2 C^{(l-1)}$. Then, because $Z_i^{(l)} = AY_i^{(l)} + \mathbf{1}_{N \times 1} b_i^{(l)}$, we obtain

$$K^{(l)} = \sigma_b^2 \mathbf{1}_{N \times N} + A \mathrm{Cov}(Y_i^{(l)}) A^T = \sigma_b^2 \mathbf{1}_{N \times N} + \sigma_w^2 A C^{(l-1)} A^T,$$

which concludes the proof.

## B.2 PROOF OF LEMMA 2

We know that $\phi(w \cdot x)$ is the feature mapping of the arc-cosine kernel. It suffices to show for different $x_1, \ldots, x_m \in S$, the $\phi(w, x_i)$'s are linearly independent.

We prove this by contradiction. Assume there exist $c_1, \ldots, c_m$ not simultaneously zero, such that

$$\sum_{i=1}^m c_i \phi(w \cdot x_i) \equiv 0. \tag{9}$$

WLOG, we further assume that $m$ is the smallest integer that satisfies (9). Let $e_1 = (1, 0, \cdots, 0)$; then $x \cdot e_1 > 0, \forall x \in S$. Assume $x_m \cdot e_1 = \min_{1 \le i \le m}\{x_i \cdot e_1\} > 0$. Then, for $1 \le i \le m - 1$,

$$\frac{x_i \cdot x_m}{x_i \cdot e_1} < \frac{1}{x_m \cdot e_1} = \frac{x_m \cdot x_m}{x_m \cdot e_1}.$$

Therefore, there exists $t$ such that

$$\frac{x_i \cdot x_m}{x_i \cdot e_1} < t < \frac{x_m \cdot x_m}{x_m \cdot e_1}, \quad 1 \le i \le m - 1.$$

Let $w = x_m - te_1$; then

$$w \cdot x_m > 0, \quad w \cdot x_i < 0, \quad 1 \le i \le m - 1.$$

Using (9) we know $c_m(w \cdot x_m) = 0$; so $c_m = 0$. Thus, $c_1, \cdots, c_{m-1}$ and $\phi(w \cdot x_1), \cdots, \phi(w \cdot x_{m-1})$ also satisfy (9), with a smaller number $m$, which forms a contradiction.

### B.3 PROOF OF THEOREM 3

From the recursive relation (3)–(4), we know $C^{(0)}$ is element-wise non-negative. Hence, $K^{(1)}$ is element-wise non-negative, $C^{(1)}$ is element-wise positive, and $K^{(2)}$ is element-wise positive.

Assume the Cholesky decomposition $K^{(2)} = LL^T$ and denote $L = (l_1, \cdots, l_n)^T$. Let $r_i = \|l_i\|_2 > 0$ and $x_i = \frac{l_i}{r_i}$. Then, $x_1 = \pm e_1$, and WLOG we assume $x_1 = e_1$. Then for each $x_i$, we know

$$x_i \cdot e_1 = \frac{l_i \cdot l_1}{r_i r_1} = \frac{K^{(2)}(i,1)}{r_i r_1} > 0$$

and each $\|x_i\|_2 = 1$. Therefore, $x_i \in S$ is defined in Lemma 2.

By the assumption that $C^{(0)}$ does not contain two linearly dependent rows, we know so are $K^{(1)}$ and $K^{(2)}$. Thus, no two $l_i$'s are linearly dependent and hence $x_i$'s are pairwise distinct. Using the universality of the arc-cosine kernel (Lemma 2), we know $C^{(2)} = g(K^{(2)})$ is positive definite. Hence, $K^{(3)}$ is positive definite. An induction argument on the layer index completes the proof.

### B.4 PROOF OF THEOREM 4

For a covariance matrix $K$, denote its standard deviation and correlation by

$$\sigma_K(x) = \sqrt{K(x,x)},$$
$$\rho_K(x,x') = \frac{K(x,x')}{\sqrt{K(x,x)K(x',x')}}.$$

From the closed-form formula (5) for the arc-cosine kernel, we have

$$\rho_{C^{(l)}}(x,x') = \frac{1}{\pi}\left(\sin\theta_{x,x'}^{(l)} + (\pi - \theta_{x,x'}^{(l)})\cos\theta_{x,x'}^{(l)}\right),$$
$$\rho_{K^{(l)}}(x,x') = \cos\theta_{x,x'}^{(l)}.$$

We define the correlation mapping from $\rho_{K^{(l)}}$ to $\rho_{C^{(l)}}$:

$$f : \cos\theta \mapsto \frac{1}{\pi}\left(\sin\theta + (\pi - \theta)\cos\theta\right), \quad \theta \in [0, \pi].$$

We first establish a few properties of $f$. A pictorial illustration of $f$ is given in Figure 5.

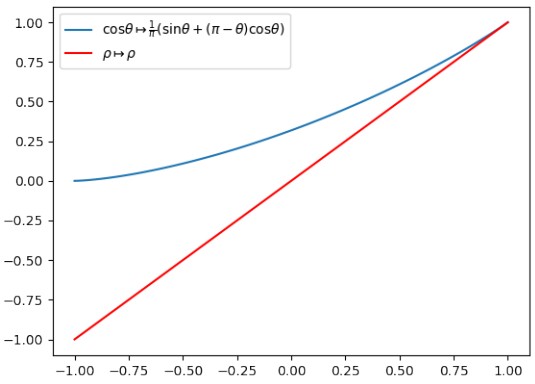

Figure 5: Correlation mapping $f$.

**Lemma 5.** *The following facts hold.*

*1. $f$ is increasing and it is a contraction mapping on $[-1, 1]$.*

2. $f(\rho) \geq \rho$ and the equality holds only when $\rho = 1$.

*Proof.* For part 1, denote $\rho = \cos\theta \in [-1, 1]$. Then,

$$f'(\rho) = \frac{\partial f(\rho)}{\partial \theta} \frac{\partial \theta}{\partial \rho} = \frac{-1}{\pi}(\pi - \theta)\sin\theta \cdot \frac{-1}{\sin\theta} = \frac{\pi - \theta}{\pi} \in [0, 1].$$

Thus, $f$ is increasing and it is a contraction mapping on $[-1, 1]$.

For part 2, we know that $f(1) = 1$. Using the contraction mapping, we know $f(\rho) \geq \rho$ for $\rho \in [-1, 1]$. Clearly, the equality holds only when $\rho = 1$. $\qquad\square$

After the correlation mapping $f$, we further consider the covariance mapping $g : K^{(l)} \mapsto C^{(l)}$ defined in (3) (and equivalently in (8)). We establish a few properties of $g$ below.

**Lemma 6.** *For positive semi-definite matrices $B$, the following facts hold.*

1. $g(B) = \frac{1}{2}D^{1/2}f(D^{-1/2}BD^{-1/2})D^{1/2}$, where $D = \text{diag}\{B\} \in \mathbb{R}^{N \times N}$ and $f$ is elementwise.

2. $g(B) \succ \frac{1}{2}B$ and $\text{diag}\{g(B)\} = \frac{1}{2}\text{diag}\{B\}$, where $\succ$ means elementwise greater.

3. $\text{tr}\left(Ag(B)A^T\right) \leq \frac{1}{2}\|A\|_2^2\text{tr}(B)$, where $\|A\|_2$ is the spectral norm of matrix $A$.

*Proof.* For part 1, it is obvious according to (5).

For part 2, using the properties of $f$ we straightforwardly obtain

$$g(B) = \frac{1}{2}D^{1/2}f(D^{-1/2}BD^{-1/2})D^{1/2} \succ \frac{1}{2}D^{1/2}(D^{-1/2}BD^{-1/2})D^{1/2} = \frac{1}{2}B.$$

For part 3, note that $g(B)$ is positive semi-definite. Using von Neumann's trace inequality,

$$\text{tr}\left(Ag(B)A^T\right) = \text{tr}\left(A^TAg(B)\right) \leq \sum_{i=1}^{N}\lambda_i(A^TA)\lambda_i(g(B)) \leq \|A^TA\|_2\sum_{i=1}^{N}\lambda_i(g(B)),$$

where the last term is equal to $\|A\|_2^2\text{tr}(g(B)) = \frac{1}{2}\|A\|_2^2\text{tr}(B)$. $\qquad\square$

Now, we are ready to prove Theorem 4.

*Proof of Theorem 4, part 1.* We focus on uniform convergence over all pairwise correlations. Denote

$$\rho_{\min}(K) = \min_{x,x'}\rho_K(x, x') \in [-1, 1].$$

To simplify notation, we temporarily let $C = C^{(l)}$. Then,

$$\begin{aligned}
K^{(l+1)}(x, x') &= \sigma_w^2\sum_{v,v'}A_{xv}A_{x'v'}C(v, v') \\
&= \sigma_w^2\sum_{v,v'}A_{xv}A_{x'v'}\sigma_C(v)\sigma_C(v')\rho_C(v, v') \\
&\geq \rho_{\min}(C) \cdot \sigma_w^2\sum_{v,v'}A_{xv}A_{x'v'}\sigma_C(v)\sigma_C(v') \\
&= \rho_{\min}(C) \cdot \sigma_w^2\left(\sum_v A_{xv}\sigma_C(v)\right)\left(\sum_{v'}A_{x'v'}\sigma_C(v')\right).
\end{aligned}$$

Meanwhile,

$$K^{(l+1)}(x, x) = \sigma_w^2 \sum_{v,v'} A_{xv} A_{xv'} C(v, v')$$

$$\leq \sigma_w^2 \sum_{v,v'} A_{xv} A_{xv'} \sigma_C(v) \sigma_C(v')$$

$$= \sigma_w^2 \left( \sum_v A_{xv} \sigma_C(v) \right)^2$$

and similarly

$$K^{(l+1)}(x', x') \leq \sigma_w^2 \left( \sum_{v'} A_{x'v'} \sigma_C(v') \right)^2.$$

Combining the above inequalities, we obtain

$$\rho(K^{(l+1)})(x, x') = \frac{K^{(l+1)}(x, x')}{\sqrt{K^{(l+1)}(x, x) K^{(l+1)}(x', x')}} \geq \rho_{\min}(C^{(l)}), \quad \forall x, x',$$

and thus

$$\rho_{\min}(K^{(l+1)}) \geq \rho_{\min}(C^{(l)}).$$

From the correlation mapping, we know

$$\rho_{\min}(C^{(l)}) = f(\rho_{\min}(K^{(l)})).$$

Therefore, by the properties of $f$,

$$\rho_{\min}(K^{(l+1)}) \geq f(\rho_{\min}(K^{(l)})) \geq \rho_{\min}(K^{(l)}).$$

Thus, $\rho_{\min}(K^{(l)}) \nearrow$ some $\rho_\infty \in [-1, 1]$. Because $\rho_\infty = f(\rho_\infty)$, we conclude that $\rho_\infty = 1$. $\quad\square$

*Proof of Theorem 4, part 2.* Consider each $h : K^{(l)} \mapsto K^{(l+1)}$:

$$K^{(l+1)} = \sigma_b^2 \mathbf{1}_{N \times N} + \sigma_w^2 A C^{(l)} A^T = \sigma_b^2 + \sigma_w^2 A g(K^{(l)}) A^T := h(K^{(l)}).$$

Let $\delta = \frac{1}{2} \sigma_w^2 \lambda^2 < 1$. Using part 3 of Lemma 6,

$$\operatorname{tr}\left(K^{(l+1)}\right) = \operatorname{tr}\left(\sigma_b^2 \mathbf{1}_{N \times N}\right) + \operatorname{tr}\left(\sigma_w^2 A g(K^{(l)}) A^T\right)$$

$$= N \sigma_b^2 + \sigma_w^2 \operatorname{tr}\left(A g(K^{(l)}) A^T\right)$$

$$\leq N \sigma_b^2 + \frac{1}{2} \sigma_w^2 \lambda^2 \operatorname{tr}\left(K^{(l)}\right)$$

$$= N \sigma_b^2 + \delta \cdot \operatorname{tr}\left(K^{(l)}\right).$$

Therefore, for sufficiently large $l$,

$$\operatorname{tr}\left(K^{(l)}\right) \leq \frac{N \sigma_b^2}{1 - \delta} + 1.$$

Because each $K^{(l)}$ is positive semi-definite, we know $K^{(l)}$ is bounded. Thus, a subsequence of $K^{(l)} \to$ some matrix. $\quad\square$

*Proof of Theorem 4, part 3.* Define $\kappa^{(l)} := K^{(l)}/c_l$. We have the following recursive relationship between $\kappa^{(l)}$ and $\kappa^{(l+1)}$:

$$\kappa^{(l+1)} = \frac{1}{c_{l+1}}(\sigma_b^2 \mathbf{1}_{N \times N} + \sigma_w^2 A g(K^{(l)}) A^T) = \frac{\sigma_b^2}{c_{l+1}} \mathbf{1}_{N \times N} + \frac{2}{\lambda^2} A g(\kappa^{(l)}) A^T.$$

Then,

$$\operatorname{tr}\left(\kappa^{(l+1)}\right) = \operatorname{tr}\left(\frac{\sigma_b^2}{c_{l+1}} \mathbf{1}_{N \times N}\right) + \frac{2}{\lambda^2} \operatorname{tr}\left(A g(\kappa^{(l)}) A^T\right) \leq \frac{N \sigma_b^2}{c_{l+1}} + \operatorname{tr}\left(\kappa^{(l)}\right),$$

where the inequality results from part 3 of Lemma 8. Therefore, $\kappa^{(l)}$ is bounded.

Using the Perron-Frobenius theorem, we know the eigenspace $V$ corresponding to $\lambda$ is one-dimensional, and there exists $v \in V$ such that $\|v\|_2 = 1$ and $v \succ 0$. Then,

$$v^T \kappa^{(l+1)} v = \tfrac{\sigma_b^2}{c_{l+1}} v^T \mathbf{1}_{N \times N} v + \tfrac{2}{\lambda^2} v^T A g(\kappa^{(l)}) A^T v \geq 2 v^T g(\kappa^{(l)}) v \geq v^T \kappa^{(l)} v.$$

Therefore, $v^T \kappa^{(l)} v$ has a limit $c > 0$.

Let $P = \mathrm{Proj}_{V^\perp} = I - \mathrm{Proj}_V = I - vv^T$. Then $PA = AP$, $\|PA\|_2 = \lambda' < \lambda$, and

$$\mathrm{tr}\left( P g(\kappa^{(l)}) P^T \right) = \mathrm{tr}\left( g(\kappa^{(l)}) \right) - v^T g(\kappa^{(l)}) v \leq \tfrac{1}{2}\mathrm{tr}\left( \kappa^{(l)} \right) - \tfrac{1}{2} v^T \kappa^{(l)} v = \tfrac{1}{2}\mathrm{tr}\left( P \kappa^{(l)} P^T \right).$$

Therefore,

$$\begin{aligned}
\mathrm{tr}\left( P \kappa^{(l+1)} P^T \right) &= \mathrm{tr}\left( \tfrac{\sigma_b^2}{c_{l+1}} P \mathbf{1}_{N \times N} P^T \right) + \tfrac{2}{\lambda^2}\mathrm{tr}\left( P A g(\kappa^{(l)}) A^T P^T \right) \\
&\leq \tfrac{N\sigma^2}{c_{l+1}} + \tfrac{2}{\lambda^2}\mathrm{tr}\left( PAP g(\kappa^{(l)}) P^T A^T P^T \right) \\
&\leq \tfrac{N\sigma^2}{c_{l+1}} + \tfrac{2}{\lambda^2}\|PA\|_2^2 \mathrm{tr}\left( P g(\kappa^{(l)}) P^T \right) \\
&\leq \tfrac{N\sigma^2}{c_{l+1}} + (\tfrac{\lambda'}{\lambda})^2 \mathrm{tr}\left( P \kappa^{(l)} P^T \right).
\end{aligned}$$

Thus, $P\kappa^{(l)} P^T \to 0$.

Combining the results, we have $\kappa^{(l)} \to cvv^T$. $\qquad\square$

## B.5 RESULTS FOR MULTI-LAYER PERCEPTRONS

Theorem 4 has parallel results for fully connected layers.

**Theorem 7.** *Let $A$ be the identity matrix. The following results hold as $l \to \infty$.*

1. *When $\sigma_w^2 < 2$ and $\sigma_b^2 > 0$, then $K^{(l)} \to q\mathbf{1}_{N \times N}$ where $q = \dfrac{\sigma_b^2}{1 - \sigma_w^2/2}$.*

2. *When $\sigma_w^2 > 2$, let $c_l = (\sigma_w^2/2)^l$; then $K^{(l)}/c_l \to vv^T$ where $v_x = \left[ \dfrac{\sigma_b^2}{\sigma_w^2/2 - 1} + K^{(0)}(x, x) \right]^{\frac{1}{2}}$.*

*Proof of Theorem 7, part 1.* Starting with the diagonal elements, we have

$$K^{(l+1)}(x, x) = \sigma_b^2 + \tfrac{\sigma_w^2}{2} K^{(l)}(x, x).$$

Therefore, $K^{(l)}(x, x) \to q = \dfrac{\sigma_b^2}{1 - \sigma_w^2/2}$.

For the off-diagonal elements, we have

$$\begin{aligned}
K^{(l+1)}(x, x') &= \sigma_b^2 + \tfrac{\sigma_w^2}{2}\sigma_{K^{(l)}}(x)\sigma_{K^{(l)}}(x') f\left( \rho_{K^{(l)}}(x, x') \right) \\
&\geq \sigma_b^2 + \tfrac{\sigma_w^2}{2}\sigma_{K^{(l)}}(x)\sigma_{K^{(l)}}(x')\rho_{K^{(l)}}(x, x') \\
&= \sigma_b^2 + \tfrac{\sigma_w^2}{2} K^{(l)}(x, x').
\end{aligned}$$

Therefore,

$$\liminf_l K^{(l+1)}(x, x') \geq \sigma_b^2 + \tfrac{\sigma_w^2}{2}\liminf_l K^{(l)}(x, x') \implies \liminf_l K^{(l)}(x, x') \geq q.$$

Meanwhile,

$$\limsup_l K^{(l)}(x, x') \leq \limsup_l \sqrt{K^{(l)}(x, x) K^{(l)}(x', x')} = q.$$

Thus, $K^{(l)}(x, x') \to q$. In conclusion, $K^{(l)} \to q\mathbf{1}_{N \times N}$. $\qquad\square$

*Proof of Theorem 7, part 2.* Again, we start with diagonal elements. Define $\kappa^{(l)} := K^{(l)}/c_l$. When $\sigma_w^2 > 2$, we have

$$\kappa^{(l+1)}(x, x) = \frac{\sigma_b^2}{(\sigma_w^2/2)^{l+1}} + \kappa^{(l)}(x, x).$$

Therefore,

$$\lim_{l \to \infty} \kappa^{(l)}(x, x) = \sum_{l=0}^{\infty} \frac{\sigma_b^2}{(\sigma_w^2/2)^{l+1}} + \kappa^{(0)}(x, x) = \frac{\sigma_b^2}{\sigma_w^2/2 - 1} + K^{(0)}(x, x) = v_x^2.$$

For the off-diagonal elements, we have

$$\kappa^{(l+1)}(x, x') = \frac{\sigma_b^2}{c_{l+1}} + \sigma_{\kappa^{(l)}}(x)\sigma_{\kappa^{(l)}}(x')f\left(\rho_{\kappa^{(l)}}(x, x')\right)$$
$$\geq \sigma_{\kappa^{(l)}}(x)\sigma_{\kappa^{(l)}}(x')\rho_{\kappa^{(l)}}(x, x')$$
$$= \kappa^{(l)}(x, x').$$

Thus, $\kappa^{(l)}(x, x')$ has a limit $c$. Taking limits on both sides, we have

$$c = v_x v_{x'} \cdot f\left(c/(v_x v_{x'})\right),$$

which suggests that $c = v_x v_{x'}$. In conclusion, $\kappa^{(l)} \to vv^T$. $\square$

## C   COMPOSING GRAPHSAGE

As before, $X$ denotes pre-activation rather post activation. A GraphSAGE layer has two weight matrices, $W_1$ and $W_2$. Hence, we use two variance parameters $\sigma_{w_1}$ and $\sigma_{w_2}$ for them, respectively. For simplicity, we focus on the mean-aggregation version of GraphSAGE, where $A$ is effectively the random-walk matrix. The updates are:

$$\text{GraphSAGE}: \quad X \leftarrow \phi(X)W_1 + A\phi(X)W_2$$
$$K \leftarrow \sigma_{w_1}^2 g(K) + \sigma_{w_2}^2 Ag(K)A^T$$
$$Q \leftarrow [\sigma_{w_1}P \quad \sigma_{w_2}AP] \quad \text{where} \quad P = \text{Chol}(g(QQ^T)).$$

Note that we do not use a bias term. If it is desired, one can easily add it to the updates following the example of GCN.

## D   EXPERIMENT DETAILS

**Datasets.** A summary is given in Table 5. The datasets Cora/Citeseer/PubMed/Reddit, with pre-defined training/validation/test splits, are downloaded from the PyTorch Geometric library (Fey & Lenssen, 2019) and used as is. The dataset ArXiv comes from the Open Graph Benchmark (Hu et al., 2020b). The datasets Chameleon/Squirrel/Crocodile come from MUSAE (Rozemberczki et al., 2021). The training/validation/test splits of the former two sets of datasets come from Geom-GCN (Pei et al., 2020), in accordance with the PyTorch Geometric library. The split for Crocodile

Table 5: Dataset summary. The edge column lists the number of directed edges.

| Dataset | Task | Nodes | Edges | Features | Train/Val/Test Ratio |
|---------|------|-------|-------|----------|---------------------|
| Cora | Classification | 2,708 | 5,429 | 1,433 | 0.05/0.18/0.37 |
| Citeseer | Classification | 3,327 | 4,732 | 3,703 | 0.04/0.15/0.30 |
| PubMed | Classification | 19,717 | 44,338 | 500 | 0.003/0.025/0.051 |
| ArXiv | Classification | 169,343 | 1,166,243 | 128 | 0.54/0.18/0.28 |
| Reddit | Classification | 232,965 | 11,606,919 | 602 | 0.66/0.10/0.24 |
| Chameleon | Regression | 2,277 | 36,101 | 3,132 | 0.48/0.32/0.20 |
| Squirrel | Regression | 5,201 | 217,073 | 3,148 | 0.48/0.32/0.20 |
| Crocodile | Regression | 11,631 | 180,020 | 13,183 | 0.48/0.32/0.20 |

is not available, so we conduct a random split with the same 0.48/0.32/0.20 proportion as that used for Chameleon and Squirrel (Rozemberczki et al., 2021).

**Dataset preprocessing.** For each graph, we treat the edges as undirected and construct a binary, symmetric adjacent matrix $\underline{A}$. Then, we apply the normalization proposed by Kipf & Welling (2017) to define $A$ used in (1). Specifically, $A = (I_N + D)^{-1/2}(I + \underline{A})(I_N + D)^{-1/2}$, where $D = \text{diag}\{\sum_j \underline{A}_{ij}\}$. For GraphSAGE and GGP, we instead use the row-normalized version $A = (I_N + D)^{-1}(I_N + \underline{A})$. No feature preprocessing is done except running the GPs on ArXiv, for which we apply a centering on the input features.

**Environment.** All experiments are conducted on a Nvidia Quadro GV100 GPU with 32GB of HBM2 memory. The code is written in Python 3.10.4 as distributed with Ubuntu 22.04 LTS. We use PyTorch 1.11.0 and PyTorch Geometric 2.1.0 with CUDA 11.3.

**Training.** For all datasets except Reddit, we train GCN using the Adam optimizer in full batch for 100 epochs. Reddit is too large for GPU computation and hence we conduct mini-batch training (with a batch size 10240) for 10 epochs.

**Hyperparameters.** For classification tasks in Table 3, the hyperparameters are set to $\sigma_b = 0.0$, $\sigma_w = 1.0$, $L = 2$, hidden $= 256$, and dropput $= 0.5$. GCN is trained with learning rate 0.01. For regression tasks, they are set to $\sigma_b = \sqrt{0.1}$, $\sigma_w = 1.0$, $L = 2$, hidden $= 256$, and dropput $= 0.5$. GCN is trained with learning rate $\sqrt{0.1}$

We choose $C^{(0)}$ to be the inner product kernel. The exception is when working with the low-rank versions, we apply PCA on the input features to ensure the number of features is smaller than the number of landmark nodes.

For Table 4: GCNII uses a 2-layer architecture and sets $\alpha = 0.1$, $\lambda = 0.5$, $\beta_l = \log(\frac{\lambda}{l} + 1)$. GIN uses $L = 2$ and $\epsilon = 0.0$. GraphSAGE uses a 2-layer architecture and sets $\sigma_{w_1} = \sqrt{0.1}$ for Reddit and $\sigma_{w_1} = 0.0$ for other datasets, and $\sigma_{w_2} = 1.0$.

The nugget $\epsilon$ is chosen using a grid search over $[10^{-3}, 10^1]$.

For GCNGP-X, GNNGP-X and RBF-X, the landmark points are chosen to be the training set for small datasets, while for large datasets (ArXiv and Reddit) the landmark points are a random 1/50 of the training set.

**Kernels.** For the **RBF** kernel,

$$K(x, x') = \exp(-\gamma \|x - x'\|_2^2),$$

where $\gamma$ is chosen using a grid search over $[10^{-2}, 10^2]$. For the **GGP** kernel,

$$K = A K_0 A^T, \quad K_0(x, x') = (x^T x' + c)^d,$$

where $c = 5.0, d = 3.0$. Note that we use only the kernel but not the robust-max likelihood nor the ELBO training proposed by Ng et al. (2018), because their code written in Python 2 is outdated and cannot be used. Note also that GGP-X in Ng et al. (2018) has a different meaning—training GGP by using additionally the 500 nodes in the validation set—while we use GGP-X to denote the Nyström approximation of GGP.

