# OpenReview forum: "Graph Neural Network-Inspired Kernels for Gaussian Processes in Semi-Supervised Learning"
_ICLR.cc/2023/Conference — ICLR 2023 poster_

### Official Review · Reviewer_SNQW · 2022-10-23

**Confidence:** 3
**Clarity, Quality, Novelty And Reproducibility:** 1. The paper is written clearly.
2. T…
**Correctness:** 4
**Technical Novelty And Significance:** 4
**Empirical Novelty And Significance:** 3
**Recommendation:** 6

**Strength And Weaknesses:**

Strength:

1. The designed GP covariance kernel is able to capture the geometry information of the graph.
2. The runtime of the designed GP is much faster than standard GCN, especially using the low-rank version. Also, its performance on all datasets is better than GCN.
3. For most of the operations in GNNs, operational counterparts for the GP exist. Therefore, the proposed GP model can be extended to many other GNNs.

Weaknesses:

1. It is interesting to see that GP versions of many SOTA GNNs achieve comparable performance with respect to their original models. However, this is only done in one dataset. It would be great if the authors can also have this comparison on other datasets in Table 4.


Questions:
1. How is $d_0$ defined in $C^{(0)}$?
2. For usual GPs, their hyperparameters (typically in the prior mean function and the covariance function) can be obtained by optimizing the log-marginal likelihood. In the proposed GP, how can these hyperparameters be optimized?
3. Can the proposed GP model be extended to the GAT family?


**Summary Of The Paper:**

The paper derives a GP model as a GCN with layer widths increased to infinity. To make the prediction more scalable, a low-rank approximation approach is proposed. Experiments are conducted to compare with other baseline models and show the GP model’s running time and scalability advantages.

**Summary Of The Review:**

The idea of deriving a GP model for infinitely wide GCN is novel. Also, the GP model can be extended to a wide range of GNNs, which shows its generality.

---

> ### Author Response · Authors · 2022-11-17
> **Response to Reviewer SNQW**
>
> **RE: Experiment on more datasets in Table 4. (We believe the reviewer means Table 5).**
>
> We have added results for other datasets to the table. These datasets in the table consistently show that the observations on GCN can be similarly extended to those on other GNNs.
>
> **RE: How is $d_0$ defined in $C^{(0)}$?**
>
> The quantity $d_0$ is the initial feature dimension of the graph nodes.
>
> **RE: Optimization of the hyperparameters.**
>
> We optimize the hyperparameters through tuning, a practice adopted by training neural networks. There are three reasons for this. First, folklore wisdom suggests that maximum likelihood yields hyperparameters suboptimal for predictions. Second, the Gaussian marginal likelihood applies only to regression but not classification (where the prior is not Gaussian). We have tried well-known approximations in the literature (such as the Laplace approximation to the posterior) but the prediction performance is less good. Third, we have also tried using numerical optimization to directly optimize the cross-entropy loss for classification. The optimization is not robust in the area where the covariance matrix is nearly singular.
>
> **RE: Can the proposed GP model be extended to the GAT family?**
>
> It is a challenge to extend the kernel derivation to GAT, because the weight matrix $W$ in each layer is used in multiple places, including the redefinition of the matrix $A$. The covariance computation cannot be simply decomposed into basic building blocks as those summarized in Table 2. A separate treatment may be needed to handle this special case.
>
> **RE: Public code.**
>
> We have written a PyTorch library for this work. We will release the code on acceptance of the paper.

---

> > ### Comment · Reviewer_SNQW · 2022-12-07
> > **Thanks for the response**
> >
> > Thanks for the authors’ responses. The additional experiments and further explanations addressed my questions and concerns.

---

### Official Review · Reviewer_okj8 · 2022-10-24

**Confidence:** 3
**Correctness:** 3
**Technical Novelty And Significance:** 4
**Empirical Novelty And Significance:** 3
**Recommendation:** 6

**Clarity, Quality, Novelty And Reproducibility:**

- The manuscript is readable.
- Technically sound.
- The proposed formulation seems novel, but some related works are missing.

**Strength And Weaknesses:**

S1. The authors derive GCN-inspired kernels by defining GP as the limit of GCN.

S2. The theoretical guarantees such as universality are thoroughly discussed.

S3. A practical inference algorithm is also presented, whose effectiveness is verified on real-world datasets.

W1. There is no discussion of important related studies, graph Gaussian processes [R1]. Can you please explain in detail the relationship with the proposed method? Please add to the baseline method if necessary.

[R1] Yin Cheng Ng et. al., Bayesian Semi-supervised Learning with Graph Gaussian Processes, NeurIPS, 2018.

**Summary Of The Paper:**

This paper defines GCN-inspired kernels by deriving GP as the limit of GCN. The authors present some theoretical results such as universality and develop a scalable posterior inference method based on low-rank approximation of covariance matrix. The effectiveness of the proposed method is demonstrated using multiple bench mark datasets.

**Summary Of The Review:**

Connection of GP and GCN is interesting. The proposed formulation is reasonable and the approximation method is beneficial. But discussion about related works (graph Gaussian processes) are missing. Then, my opinion is "weak accept" for now.

---

> ### Author Response · Authors · 2022-11-17
> **Response to Reviewer okj8**
>
> **RE: Discussion on graph Gaussian processes.**
>
> Thank you for suggesting this paper. We have included a short discussion of it in the Related Work section. Essentially, GGP uses a kernel $AKA^T$, where A is the random-walk matrix and K is any suitable base kernel. This kernel corresponds to a 1-layer (i.e., without a hidden layer) GNN. Our kernel is more general, as it can be constructed from any number of layers, as well as from many GNN building blocks. Another difference is that GGP uses a robust-max likelihood rather than a Gaussian likelihood, as we do for simplicity. As a result, together with the use of induced points for scalability, GGP inference is made by maximizing the ELBO rather than computing a closed-form formula.
>
> **RE: Add graph Gaussian processes as a baseline.**
>
> Despite the difference, we have conducted an experiment and added the results to Table 4. For GGP, the authors’ code is outdated (written in Python 2) and cannot be adapted to the current TensorFlow/Pytorch framework. For experimentation, we take the kernel definition but remove the robust-max likelihood and the ELBO training, such that we can adapt to our software framework and obtain the results as baseline. The results of the GGP kernel are generally inferior to those of our kernel; moreover, the performance gap is greater for regression than for classification.

---

> > ### Comment · Reviewer_okj8 · 2022-12-02
> > **Thanks for your reply**
> >
> > Additional discussion and experiments are helpful and clearly addressed to my concerns. So, my opinion tends to acceptance.

---

### Official Review · Reviewer_XBR8 · 2022-10-25

**Confidence:** 3
**Clarity, Quality, Novelty And Reproducibility:** The paper is cleary written and well …
**Correctness:** 4
**Technical Novelty And Significance:** 3
**Empirical Novelty And Significance:** Not applicable
**Recommendation:** 5

**Strength And Weaknesses:**

Overall, the paper is well motivated. The use of a real data set is a huge strength of the paper. In particular, I would also like to commend the authors on the crystal clear exposition and writing throughout the paper. The proposed method and theory are clearly presented and easy to follow.

My main concern is the originality of establishing connection between GCNs and GPs. I have seen a number of previous studies which reached a similar conclusion. For example:
- https://arxiv.org/pdf/1711.00165.pdf
- https://arxiv.org/pdf/2002.12168.pdf
- https://proceedings.neurips.cc/paper/2019/hash/5e69fda38cda2060819766569fd93aa5-Abstract.html

It would be better for the authors to clarify the differences and highlight their main theoretical & methodological contributions compared to the existing studies. Also, the low rank approximation for scalable computation hinges on the Nystrom approximation of matrix K, which is widely-used in the GP literature. Therefore, in my opinion, the algorithmic contributions are incremental (or limited).

Given that I believe the authors would need to effectively address the above questions and the totality and generality of the contributions to be not significant enough to warrant publication at this stage.
I hope the authors may find these high-level comments helpful in thinking about the next steps for their paper.

**Summary Of The Paper:**

The goal of this study is to attack the semi-supervised learning problem for graph-structured data. The paper focus on bridging the gap between GNNs and the limiting GPs by deriving the covariance kernel that incorporates the graph inductive bias as GNNs do.  More specifically, the paper has shown that the GP can be derived as a limit of the GCN when the layer widths tend to infinity and demonstrated the kernel universality and the limiting behavior in depth. In addition, the paper also develops an efficient computational algorithm for posterior inference on large-scale data.

**Summary Of The Review:**

See above

---

> ### Author Response · Authors · 2022-11-17
> **Response to Reviewer XBR8**
>
> **RE: Originality and contribution.**
>
> This work presents several theoretical results, a novel computational method, and supportive empirical findings, which we would argue are important contributions that advance the knowledge of the field. We have summarized these contributions in the Introduction section, but please allow us to elaborate further, with discussions of the papers you mentioned, to elucidate the originality.
>
> - We establish the connection between GP and GCN and study the kernel universality and its limiting behavior in depth. While prior works establish similar connections, including the first and the third papers you mentioned, they are not for graphs. It remains to explore for the graph case what are the kernel formulas and the kernel properties. The second (unpublished) paper does give kernel formulas, but no properties are studied. The literature has yet to study the kernel properties for graphs. We study the kernel universality and the limiting behavior in depth. We also propose scalable computations for the concerned kernels, which have yet to be studied by the literature as well.
>
> - We propose a computational procedure to compute a low-rank approximation of the covariance matrix and to address scalability. The scalability challenge is not addressed in any of the papers you mentioned. There is a subtlety between usual neural networks and graph neural networks for this issue. For usual neural networks, the covariance for a pair of points can be computed recursively by using this pair only. In other words, the entries of the final covariance matrix can be computed “individually,” which renders low-rank approximations (such as using Nystrom or induced points) straightforward. However, for graphs, the entries can no longer be computed independently. It is nontrivial to derive a low-rank recursion. Our derivation is a substantial contribution.
>
> - We present a procedure to compose covariance kernels for a family of GNNs, beyond GCN. This has not been discussed at all in the second paper and the literature. An additional value to this contribution is that the composability applies to not only the covariance kernel but also the low-rank approximation. This benefit originates from the novelty and it resolves the scalability challenge in practice.
>
> - We report comprehensive experiment results. In particular, we demonstrate that GP can produce competitive results by using computational time that is orders of magnitude lower than training a GNN. In the literature, such attractive timing results are rarely seen. We can achieve these results owing to the nontrivial kernel approximation.
>
> During this rebuttal, we realized that we still needed some references and discussions of your mentioned papers in the initial submission. We have cited the missing references and discussed them in appropriate places in the updated version of the paper.
>
> **RE: Low-rank approximation (Nystrom).**
>
> We respectfully disagree that our contribution is limited because of the use of the Nystrom method, which has been widely used in the literature. Not only is the usage of this technique different in our context, but also the effort to make it work is nontrivial. A typical use of the Nystrom approximation in GP literature starts with a portion of the kernel matrix and computes the approximation. We cannot do so because the kernel matrix is recursively computed. Even if only some entries of the kernel matrix are needed to compute Nystrom, these entries are computed recursively, such that in each recursion, the full kernel matrix is needed because of the existence of the graph adjacency matrix $A$. Such is a critical distinction between our work and other works that deal with other neural networks using Nystrom.
>
> To obtain a low-rank approximation of the final kernel matrix feasibly, care is needed not to explode the time and memory costs. We make this important and nontrivial contribution. The solution we propose is to take an approximation in each recursion step. We develop the recursive formulas, as summarized in Algorithm 1, and extend them to other graph neural networks and their building blocks listed in Table 2. We believe these nontrivial developments contribute intellectually to the literature.

---

### Official Review · Reviewer_svk4 · 2022-10-27

**Confidence:** 3
**Clarity, Quality, Novelty And Reproducibility:** The paper is mostly well-organized. T…
**Correctness:** 3
**Technical Novelty And Significance:** 3
**Empirical Novelty And Significance:** 2
**Recommendation:** 6

**Strength And Weaknesses:**

Strength:

1. The idea is interesting and somewhat novel.
2. The manuscript is clear and easy to read.
3. The experiments validate the effectiveness of the proposed method.

Weaknesses:

1. Why does integrating graph neural network structure into GP framework improve the performance compared with GNN? In my opinion, one of the main advantages of GP is the use of kernel functions like RBF, which can approximate the nodes of infinite element network, and the kernel functions used here are still based on the inner product of mapping functions.

2. The authors emphasize that the proposed method is suitable for semi-supervised scenarios, but does not give the reason and the specific algorithm of semi-supervised learning. If it is supervised learning, whether the proposed method still has the same advantage?

3. Can the authors explain the trend of the curves in Figure 3?



**Summary Of The Paper:**

In this work, the graph neural network structure is integrated into the GP framework to construct a kernel function with graph structure, and the low-rank approximation method is used to deduce the efficient training and inference method of GP, aiming to improve the performance and efficiency of graph structure data classification and regression.

**Summary Of The Review:**

A novel idea and implementation of combing GP and graph neural network.

---

> ### Author Response · Authors · 2022-11-17
> **Response to Reviewer svk4**
>
> **RE: Why does integrating graph neural network structure into GP framework improve the performance compared with GNN?**
>
> From the existing literature, we know that for a usual neural network (e.g., a feed-forward network), its limiting behavior when the layer width tends to infinity is a random function following a Gaussian prior (i.e., a GP). The kernel of this GP, similar to RBF, is the inner product of feature maps/feature functions under an infinitely dimensional vector space. Our graph neural network case is of no difference from such. We derive the recursive formulas for the kernel.
>
> While empirical evaluations indicate that GNNGP is rather competitive, more importantly, we believe that our work strengthens the theoretical understanding of GNNs, through taking the Bayesian perspective and analyzing the limiting behavior. While the popularity of deep learning is largely driven by outstanding empirical performance, we also note that neural networks go way beyond tabular data (e.g., token sequences and graphs), which is a flexibility that typical GPs lack. We borrow the innovations on graph neural networks to GPs and improve their capabilities. One obstacle to this end is the $O(N^3)$ cost; we develop an approximation that is similarly effective but significantly faster than GNNs.
>
> **RE: Comment “The authors emphasize that the proposed method is suitable for semi-supervised scenarios, but does not give the reason and the specific algorithm of semi-supervised learning.”**
>
> In this work, we develop GP kernels inspired by GNNs and use these GPs to perform classification under a semi-supervised setting. The algorithm reuses the straightforward, closed-form formula $\hat{y}\_{\*} = K\_{\*} (K+\epsilon I)^{-1}y$ for a GP. This formula is well known, but the new development is the computation of the kernel matrix $K$, which Algorithm 1 summarizes.
>
> A reason why our GP works well is attributed to the fact that the kernel matrix $K$ is the limiting covariance matrix when the layer width of the GNN tends to infinity. As such, it inherits the strong empirical performance of GNNs, whose architecture design captures the inductive bias of a graph by using the graph adjacency matrix.
>
> **RE: If it is supervised learning, whether the proposed method still has the same advantage?**
>
> Node classification belongs to semi-supervised settings because a graph is given where some nodes are labeled while others are not. Semi-supervised learning signifies that the learning uses not only the labeled data but also the unlabeled data. Specifically, the prediction of a training node comes from not only the information of itself but also that of the neighboring nodes, some of which may be unlabeled.
>
> Supervised learning does not seem to be the right setting for node classification unless the graph connectivity information is not used and individual nodes are treated in isolation. Some of the early papers on graph neural networks (including GCN) conducted experiments to show that ignoring the graph information does not yield competitive performance.
>
> **RE: The trend of the curves in Figure 3.**
>
> Thank you for the request for clarification. The curves for GCN and GCNII agree well with the knowledge developed by the GNN literature: GCN suffers depth because of oversmoothing, while GCNII mitigates this problem by using skip and residual connections. Regarding GP, the theory (Theorem 4) suggests that the kernel matrix tends towards a rank-1 matrix, which may work less well than a full-rank matrix. Using a nugget (which models data noise) may mitigate the problem. The trends of GCNGP and GCNIIGP in Figure 3 have yet to exhibit deterioration of performance; it may take more layers before one sees the performance drop.

---

### Public Comment · ~Benedek_Andras_Rozemberczki1 · 2022-11-05
**Misattribution of datasets**

The paper misattributed the authorship of the Chameleons and Squirrels datasets. These datasets were proposed in this ICLR submission:

https://openreview.net/forum?id=HJxiMAVtPH

The Pei et al. paper cited by the authors took the Squirrel and Chameleons datasets and used those for benchmarking, but had nothing to do with the creation of the datasets. The correct citation for the paper which proposed the datasets is:

```bibtex
>@article{musae,
          author = {Rozemberczki, Benedek and Allen, Carl and Sarkar, Rik},
          title = {{Multi-Scale Attributed Node Embedding}},
          journal = {Journal of Complex Networks},
          volume = {9},
          number = {2},
          year = {2021},
}
```

---

> ### Author Response · Authors · 2022-11-17
> **Citation added**
>
> Thank you for noting the reference. We inserted the citation and in Appendix B expanded the presentation of the details on the use of the datasets.

---

### Author Response · Authors · 2022-11-17
**Summary of the rebuttal**

We thank all reviewers for the feedback! Thank you for confirming that the paper is well presented and the work is novel and interesting.

We updated the paper slightly, inserting missing references and discussions. We also added columns to Tables 4 and 5, one for additional baselines and the other for additional datasets.

While most reviewers suggest acceptance, we reiterate the value of this paper in the [response to reviewer XBR8](https://openreview.net/forum?id=flap0Bo6TK_&noteId=CSQFmue5x2), to raise awareness of our contributions to the literature.

---

### Decision · Program_Chairs · 2023-01-20

**Decision:**

Accept: poster

**Justification For Why Not Higher Score:**

The main concern is still that the proposed connection between GNNs and GPs has been studied in the previous work. The authors didn't mention this previous work at all in their initial submission. This harms the originality of the paper.


**Justification For Why Not Lower Score:**

Apart from the originality issue, the paper is well-written, addresses several interesting and critical issues in GNN-inspired kernels, and makes it scalable and generalizable to several interesting members of the GNN family. These are significant contribution to the GP community.

**Metareview: Summary, Strengths And Weaknesses:**

The paper bridges the gap between GNNs and GPs by deriving the covariance kernel that incorporates the graph inductive bias as GNNs do. More specifically, the paper has shown that the GP can be derived as a limit of the GCN when the layer widths tend to infinity and demonstrated the kernel universality and the limiting behavior in depth. Despite an earlier paper which has made the same connection between GNN and GP, this paper provides some interesting theoretical analyses and a scalable inference method which is critical for GPs with large-scale kernels. The proposed GP model can be extended to many other GNNs.

We think this paper has made a good contribution to the GP community. We all agree that the authors need to do a bit more thorough investigation into the related works, which we hope can be addressed in the final version.






**Note From Pc:**

if the above contains the word "oral" or "spotlight" please see: "oral" presentation means -> notable-top-5% and "spotlight" means -> notable-top-25%. As stated in our emails, we are disassociating presentation type from AC recommendations

**Summary Of Ac-Reviewer Meeting:**

We discussed over the major concern from the 5-score reviewer. His main concern is the paper didn't mention a previous work which is very similar to this paper. They both derive a GP model from infinitely wide GNN. We listed the difference between these two papers and concluded that this paper has more interesting analyses and provides a very practical and scalable inference method. These new components form a significant contribution and make the GNN-inspired kernel more applicable to large-scale applications and diverse GNN models.